# Simulating Scenarios of Future Intra-Urban Land-Use Expansion Based on the Neural Network–Markov Model: A Case Study of Lusaka, Zambia

**Matamyo Simwanda** [1,*]🆔**, Yuji Murayama** [2]🆔**, Darius Phiri** [1]🆔**, Vincent R. Nyirenda** [3]🆔 **and Manjula Ranagalage** [2,4]🆔

1 Department of Plant and Environmental Sciences, School of Natural Resources, Copperbelt University, Kitwe 10101, Zambia; dariusphiri@rocketmail.com
2 Faculty of Life and Environmental Sciences, University of Tsukuba, 1-1-1 Tennodai, Tsukuba, Ibaraki 305-8572, Japan; mura@geoenv.tsukuba.ac.jp (Y.M.); manjularanagalage@gmail.com (M.R.)
3 Department of Zoology and Aquatic Sciences, School of Natural Resources, Copperbelt University, Kitwe 10101, Zambia; vincent.nyirenda@cbu.ac.zm
4 Department of Environmental Management, Faculty of Social Sciences and Humanities, Rajarata University of Sri Lanka, Mihintale 50300, Sri Lanka
* Correspondence: matamyo.simwanda@cbu.ac.zm

**Abstract:** Forecasting scenarios of future intra-urban land-use (intra-urban-LU) expansion can help to curb the historically unplanned urbanization in cities in sub-Saharan Africa (SSA) and promote urban sustainability. In this study, we applied the neural network–Markov model to simulate scenarios of future intra-urban-LU expansion in Lusaka city, Zambia. Data derived from remote sensing (RS) and geographic information system (GIS) techniques including urban-LU maps (from 2000, 2005, 2010, and 2015) and selected driver variables, were used to calibrate and validate the model. We then simulated urban-LU expansion for three scenarios (business as usual/status quo, environmental conservation and protection, and strategic urban planning) to explore alternatives for attaining urban sustainability by 2030. The results revealed that Lusaka had experienced rapid urban expansion dominated by informal settlements. Scenario analysis results suggest that a business-as-usual setup is perilous, as it signals an escalating problem of unplanned settlements. The environmental conservation and protection scenario is insufficient, as most of the green spaces and forests have been depleted. The strategic urban planning scenario has the potential for attaining urban sustainability, as it predicts sufficient control of unplanned settlement expansion and protection of green spaces and forests. The study proffers guidance for strategic policy directions and creating a planning vision.

**Keywords:** intra-urban land use; artificial neural network; Markov model; modeling and simulation; scenario analysis; Lusaka

## 1. Introduction

Sustainable urban development in Africa has mainly been hindered by rapid unplanned and uncontrolled population growth and urbanization [1,2]. Over the last two decades, Africa has experienced one of the fastest urbanization rates in the world, with sub-Saharan Africa (SSA) being at the fore [3]. While most of Africa is still essentially rural, rapid human population growth and urbanization have occurred in the urban capitals that dominate their respective national economies [4]. Amidst the rapid urbanization, there has been low economic development and growth in many SSA cities [5,6]. Therefore, many cities are not capable of coping with the ever-increasing demand for basic infrastructure and services, such as housing, water and sanitation, electricity, waste disposal, employment, education, and health services [7]. These phenomena have been seen in most SSA cities emerging as unplanned cities dominated by overcrowded informal settlements haphazardly

located close to urban growth centers, such as the central business district and other industrial and commercial areas [8,9]. Consequently, ecological and environmental conditions in these cities are under continuous threat.

The challenges of urbanization in SSA are worsened due to the limited availability of data resources, useful tools, and proficiency [10], which are crucial for monitoring changes in the urban landscape. Monitoring urban land use (urban-LU) changes requires understanding and predicting the dynamic social–ecological processes and spatial patterns of urbanization at different periods [11]. In recent years, spatially explicit land-use change (LUC) models that simulate and predict future land use have become widely recognized as effective tools for monitoring changes in the urban landscape. LUC models are used to understand the causes and consequences of land-use dynamics [12], as well as explore future scenarios to support urban land-use planning and policy [13]. Therefore, LUC models present an opportunity for SSA cities to envision different future urban-LU expansion scenarios aimed at curbing the historic unplanned urban growth.

Notwithstanding the vast literature on LUC modeling, local and regional context-based models remain imperative. The interaction dynamics of urban-LU drivers (i.e., biophysical, socioeconomic, political, cultural, etc.) and the complexity of urban landscapes vary among different geographical regions. This makes it illogical to apply the same LUC models among other localities. For example, some of the notable LUC modeling (e.g., [14–16]) has been conducted in developed countries, which are characterized by highly developed urban environments and well-planned urban-LU systems. Applying these LUC models in SSA cities where urban development is mostly unplanned may present incorrect results, as the drivers may be very different.

Besides, most LUC models focus on simulating all the intra-urban-LUs (e.g., residential, industrial, commercial, public, etc.) as an individual urban/built-up category [11,13,17,18]. In the context of SSA cities, these generalized simulation approaches to the urban area can lead to broad and nonspecific pieces of advice that are not very beneficial to African urban planners and policymakers. Scaling down to intra-urban-LUs enables researchers and urban planners to understand better the structure and functioning of a city based on more solid knowledge of its inner land-use dynamics [19]. Intra-urban simulations aim to comprehend the intrinsic urban evolution through intra-urban-LU maps, urban socioeconomic and political attributes, and spatial analysis tools [20]. As such, intra-urban simulations do not only show the spatiotemporal urban-LU change processes but also explain how these trends can provide essential information for urban planners to begin possible future spatial configuration and urban revitalization [20,21]. For example, trends of intra-urban-LU dynamics can inform urban planners of areas needing urban refurbishments, improvement of public services and environmental quality, installation of urban equipment, and revitalization of the commercial and residential sectors [21].

A few recent studies have scaled down to intra-urban-LU simulations [19–24]. Cellular automata (CA) has been the most frequently applied modeling technique in these studies. However, CA models have several shortcomings in simulating intra-urban-LU changes [20]. CA models assume that urban-LU development depends on neighborhood characteristics in a particular location [25]. CA models are also strictly governed by universally applied transitional rules that rely on ad hoc definitions and heuristics, which make their calibration and validation challenging [26,27]. As a result, these models do not explicitly consider the underlying drivers [16] required for accurate intra-urban-LU simulations [20]. The accuracy of CA models is thus affected when applied to complex intra-urban-LU systems with nonlinear and unbalanced development [22]. To overcome these limitations, other studies simulating urban-LU have suggested combining CA with other applications, such as random forests (RF) [20], artificial neural networks (ANNs) [19], agent-based modeling [28], and Markov chains [23].

The mixed development of unplanned and planned intra-urban-LUs in most SSA cities displays a disordered spatial pattern suggesting a complex nonlinear and unbalanced development overtime. In the SSA city of Lusaka, Zambia, for example, Simwanda and

Murayama (2018) [8] used binary logistic regression to model the spatial dependency of urban-LUs and found that the growth of unplanned urban-LUs does not follow neighborhood characteristics; instead, they follow a spontaneous and nonlinear growth process that is either spatially dependent on the growth of planned urban-LUs (e.g., commercial and industrial areas), or is just random, depending on the availability of unoccupied land. Simwanda, Murayama, and Ranagalage (2020) [29] also used analytic network process (ANP) modeling to assess the drivers of urban-LU changes in Lusaka and concluded that the mixed unplanned and planned development of urban-LUs was driven by complex interactions of socioeconomic, population, and political factors. Therefore, the simulation of intra-urban-LUs in SSA cities requires methods that can handle the complex nonlinear interrelationships between LUC and their drivers.

Hence, in this study, we applied the neural network–Markov model to simulate scenarios of future intra-urban-LU expansion in Lusaka city, Zambia. Artificial neural networks (ANNs) have been proven to be successful in simulating LUCs in many urban settings with a high degree of complexity [19,30]. The advantages of the ANN approach (see details in [31–33]) include capabilities to handle nonlinear functions, perform model-free function estimation, learn from data relationships that are not otherwise known, and generalize to unseen situations. Therefore, ANNs make powerful tools for intra-urban-LU modeling, especially when the underlying complex relationships in data (e.g., unplanned vs. planned intra-urban-LUs in SSA cities) are unknown [33]. On the other hand, Markov models are frequently used as excellent projectors of future LUCs. Markov is a stochastic model designed to describe the probability of one state changing into another [34]. The Markov model not only explains the quantification of conversion states between the land-use types but can also reveal the transfer rate among different land-use types [35]. Therefore, combining the ANN and Markov models is a robust approach that can be used to successfully model and simulate future intra-urban-LU expansion.

To the best of our knowledge, there are no studies that have simulated future scenarios of the mixed unplanned and planned expansion of intra-urban-LUs in the context of SSA cities. Past studies have concentrated on either specifically mapping and monitoring [36–38] or assessing the drivers [25,29] of unplanned development. Karolien et al. [17] carried out a scenario-based analysis of urban growth in Kampala, Uganda, but did not scale down to the intra-urban-LU level. Our study uses data derived from RS and GIS techniques, including intra-urban-LU maps (from 2000, 2005, 2010, and 2015), and selected biophysical, neighborhood, social, and economic explanatory variables to develop, calibrate, and validate a neural network–Markov model to simulate the expansion of unplanned and planned intra-urban-LUs in Lusaka, Zambia (2000–2030). Unplanned development refers to informal settlements and all other areas developed without proper authorization from the local planning office (i.e., areas that are not in compliance with planning regulations) and vice versa. Our study considers three scenarios of intra-urban-LU expansion: (1) business as usual; (2) environmental conservation and protection; and (3) strategic urban planning, for the years 2020, 2025, and 2030. The scenarios are used to explore different alternatives for achieving future urban sustainability in Lusaka, as a case study for SSA cities. Our study is also aimed at contributing to the efforts for attaining the envisioned 2030 United Nations (UN) Global Sustainable Development Goals (SDGs), particularly SDG No. 11 (Sustainable Cities and Communities) [39].

## 2. Materials and Methods

### 2.1. Study Area

Lusaka city is located in the Lusaka Province of Zambia (Figure 1). Its central geographical coordinates are 15°24′24″ S and 28°17′13″ E (Figure 1). It lies on a plateau in the south and west with an altitude of 1200m above sea level (asl) that increases gently to 1300 m asl toward the north and east. Lusaka is the country's provincial headquarters and capital city. The city is also the political, cultural, and economic center of the country and home of the central government. As such, many institutional, commercial, and industrial activities are concentrated in Lusaka.

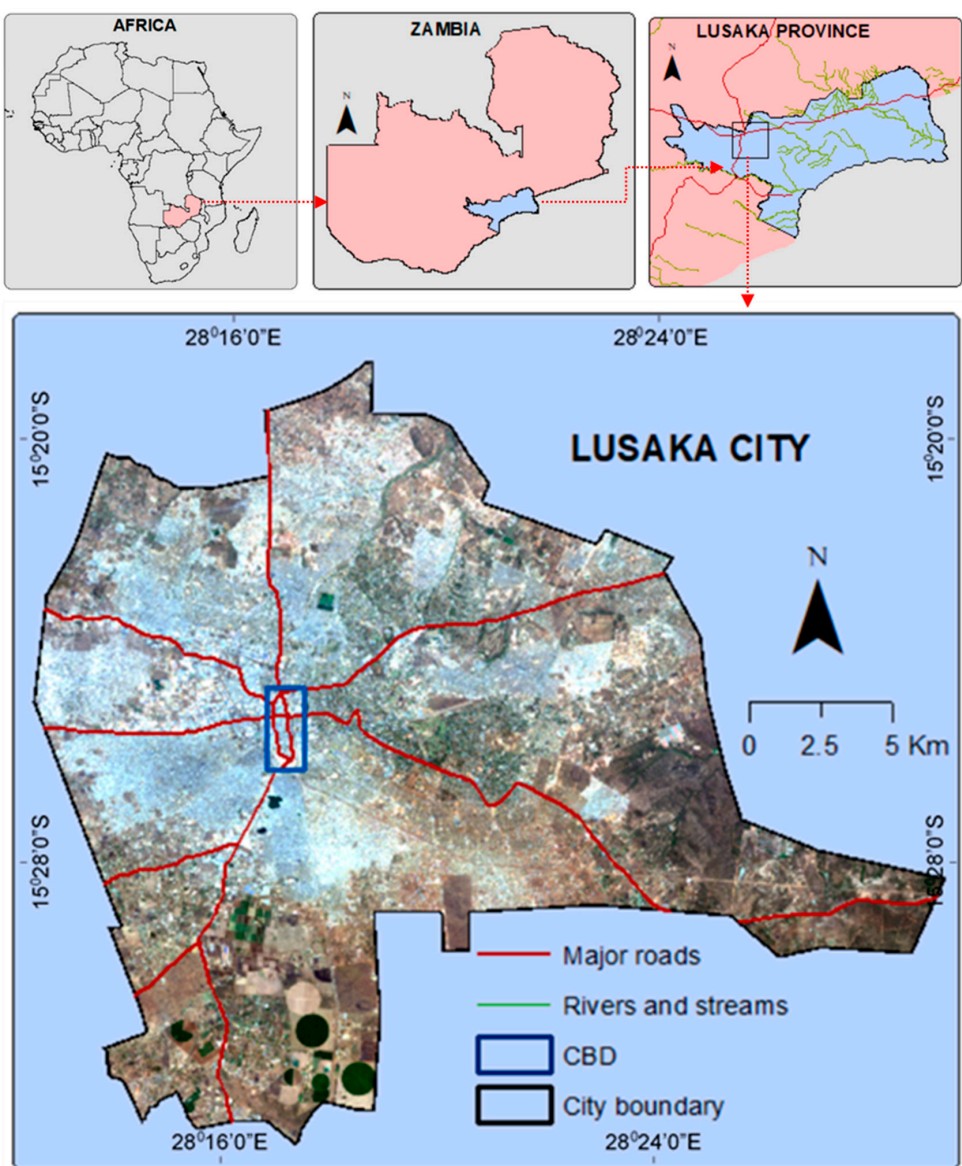

**Figure 1.** Study area: Lusaka city, located in the Lusaka Province of Zambia.

Over the last three decades, Lusaka city has expanded rapidly both in population size [40] and spatial extent [38]. The population increased from about 120,000 in 1963 to about 1.7 million in 2010 [40,41], and is currently estimated to be over 2 million. Lusaka's spatial extent expanded to 420 km$^2$ from 2.6 km$^2$ when it was initially established in 1935. Rapid urbanization has exerted pressure on land administration and delivery of social services, as well as future urban planning for sustainable development.

### 2.2. Data and Intra-Urban-LU Classification

The data used in this study included four intra-urban-LU maps for 2000, 2005, 2010, and 2015). The intra-urban-LU maps for 2000 and 2010 were obtained from [38], while the intra-urban-LU maps for 2005 and 2015 were processed in this study. Of note, an attempt was made to produce intra-urban-LU maps prior to the year 2000; however, some intra-urban-LU classes could not be modeled, as they were almost nonexistent at that time point, and thus the year 2000 was chosen as the initial year. We also could not produce intra-urban-LU maps later than 2015, as the classification methods used were highly dependent on cadastral data, which was only available up to 2015.

We used the method proposed in [38]: an integrated approach of RS and GIS techniques developed in view of the inherent challenges of intra-urban-LU classification in SSA cities (i.e., spatial complexity and spectral confusion; lack of high-resolution satellite imagery, and the limited applicability of available intra-urban-LU classification methods). To produce the intra-urban-LU maps, first we extracted the built-up land from Landsat TM/+ETM data using a combination of pixel-based and object-based classification techniques, plus post-classification image control for accuracy purposes. Second, we created land parcels as homogenous units separating intra-urban-LU classes in the study area using road-network data. Then, we classified the parcels into six separate intra-urban-LU classes using a combined detail of cadastral maps, land-use data, and high-resolution Google Earth imagery. The parcels were then used to divide the initially extracted built-up land into the six intra-urban-LU classes (Table 1). The residential intra-urban-LU classes were further separated by residential density (RD), measured in dwelling units (du) per square kilometres (km$^2$); and population density (PD), measured in people per km$^2$. A detailed description of the intra-urban-LU mapping approach and accuracy assessment can be found in [38]. The overall accuracy values of the intra-urban-LU maps for 2000, 2005, 2010, and 2015 were 91.3%, 88.9%, 93.0%, and 92.1%, respectively. Thus, the overall accuracies of the intra-urban-LU maps obtained from [38] (2000 and 2010) and those produced in this study (2005 and 2015) were relatively similar, as they all exceeded the minimum standard of 85%.

**Table 1.** Descriptions of the intra-urban-LU classes [38].

| No. | Intra-Urban-LU Class | Description |
|---|---|---|
| 1 | Unplanned High-Density Residential (UHDR) | Unplanned [1] residential areas comprising dense informal settlements with very high population density (RD > 2000 du/km$^2$ and PD of 12,000–28,000 people/km$^2$) |
| 2 | Unplanned Low-Density Residential (ULDR) | All unplanned residential areas with medium- to high-cost housing and low population density (RD < 2000 du/km$^2$ and PD of 400–2000 people/km$^2$). |
| 3 | Planned Medium-/High-Density Residential (PMHDR) | Planned [2] residential areas with low- to medium-cost houses with high population density (RD > 2000 du/km$^2$ and PD of 2000–22,000 people/km$^2$). |
| 4 | Planned Low-Density Residential (PLDR) | Planned areas with high-cost houses and low population density RD < 2000 du/km$^2$ and PD of 400–2000 people/km$^2$). |
| 5 | Commercial and Industrial (CMI) [3] | Areas comprising general retail, shopping malls, markets, hotels, financial services, manufacturing, warehousing, quarrying, and commercial agriculture facilities |
| 6 | Public Institutions and Service (PIS) [4] | Areas comprising education and health facilities, religious institutions, government and administration houses, municipal utilities, transportation terminals, and aviation facilities |

Note: [1] Unplanned refers to all residential areas that developed without proper authorization as declared by the local planning authority. These areas lack basic services and are mainly characterized by sparse road networks, poor access to electricity, no water or sanitation, and poor access to social amenities like public schools, health centers, and public recreational open spaces. [2] Planned refers to all areas that developed with proper authorization from the local planning office. These areas generally have functional basic services, including paved roads and municipal water and sanitation services. [3] CMI refers to areas for economic activities. [4] PIS refers to areas for public social services (see Table 2 below as well).

Ancillary spatial data were also collected to aid in modeling and simulation. The ancillary data included the latest city administrative boundary, a 1985 topographic map, a 2003 partial QuickBird satellite image (0.6 m spatial resolution), and the Shuttle Radar Topography Mission (SRTM) 30 m resolution digital elevation model (DEM) for Lusaka downloaded from the US Geological Survey website. The Lusaka urban development plan (2010–2030) was also obtained to assist in scenario development for future intra-urban-LU simulation.

**Table 2.** Representation characteristics of driver variables (refer to Figure 3 for abbreviations).

| No. | Driver Variable | Representation Characteristics | Nature of Representation (Influence of) |
|---|---|---|---|
| 1 | UHDR | 1. Physical<br>2. Neighborhood<br>3. Social/economic<br>4. Policy/planning | 1. Informal settlements with highest number of du/km$^2$<br>2. Highest population density areas<br>3. Residential areas with high poverty levels and poor public services<br>4. Spatial planning and policy challenge |
| 2 | ULDR | 1. Physical<br>2. Neighborhood<br>3. Social<br>4. Policy/planning | 1. Unplanned residential areas with low number of du/km$^2$<br>2. Low population density areas<br>3. Residential areas with poor access to public services<br>4. Spatial planning and policy challenges |
| 3 | PMHDR | 1. Physical<br>2. Neighborhood<br>3. Social<br>4. Policy/planning | 1. Planned residential areas with medium-high number of du/km$^2$<br>2. Medium-high population density areas<br>3. Residential areas with adequate access to public services<br>4. Spatial planning and policy implementations |
| 4 | PLDR | 1. Physical<br>2. Neighborhood<br>3. Social<br>4. Policy/planning | 1. Planned residential areas with low number of du/km$^2$<br>2. Low population density areas<br>3. Residential areas with adequate access to public services<br>4. Spatial planning and policy implementations |
| 5 | CMI | 1. Physical<br>2. Economic | 1. All commercial and industrial areas including general retail, markets, hotels, financial services, manufacturing, warehousing, quarrying, and commercial agriculture facilities<br>2. Accessibility to economic activities |
| 6 | PIS | 1. Physical<br>2. Social | 1. All public institutions and service areas including education and health facilities, religious institutions, government and administration houses, municipal utilities, transportation terminals, and aviation facilities<br>2. Accessibility to public social services |
| 7 | Major roads | 1. Social/economic | 1. Accessibility to transportation |

**Table 2.** *Cont.*

| No. | Driver Variable | Representation Characteristics | | Nature of Representation (Influence of) | |
|---|---|---|---|---|---|
| 8 | CBD | 1. | Economic | 1. | Accessibility to city center with major economic activities |
| 9 | DEM | 1. | Physical | 1. | Topography effects |
| 10 | Slope | 1. | Physical | 1. | Topography effects |
| 11 | Forest | 1. | Biophysical | 1. | Accessibility to forests |
| 12 | Agriculture | 1. | Biophysical | 1. | Accessibility to agriculture/cultivation/crop lands |
| 13 | Water | 1. | Biophysical | 1. | Accessibility to water resources: streams, rivers, and lakes |

### 2.3. Modeling Intra-Urban-LU Expansion

In this study, we used the land-change modeler (LCM) provided in the TerrSet Geospatial Monitoring and Modeling software [42] to implement the neural network–Markov model. The LCM has a set of tools for analyzing land use/land cover, empirically modeling its relationship to explanatory variables and simulating scenarios of future land-use/land-cover changes [42]. The LCM is one of the leading modeling systems, and has been widely applied in studies involving modeling and simulating urban expansion (e.g., [10,11,43–46].

The development and implementation of the neural network–Markov model in this study involved four steps: (1) selection of model explanatory (or driver) variables; (2) transition-potential modeling and simulation; (3) model validation; and (4) scenario development and simulation of future intra-urban-LU expansion. The modeling framework applied in this study is presented in Figure 2.

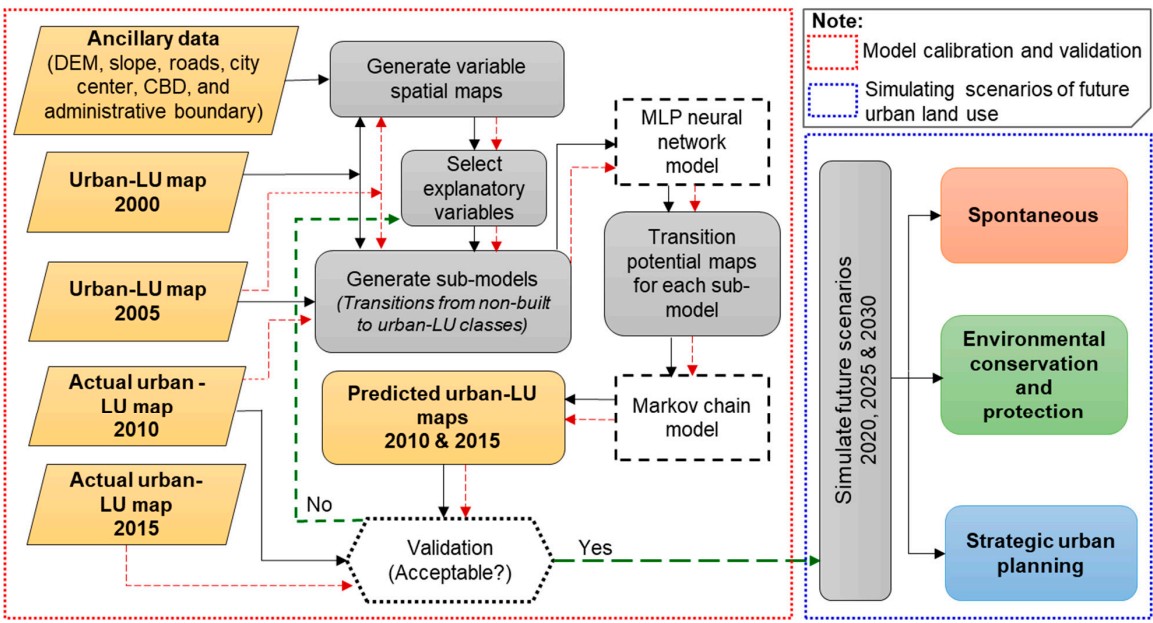

**Figure 2.** Modeling framework for simulating scenarios of future intra-urban-LU expansion of Lusaka city, Zambia. Note: intra-urban-LU maps for 2000 and 2005 were used for calibration and simulation to 2010 (black arrows), while intra-urban-LU maps for 2005 and 2010 were used for calibration and simulation to 2015 (dashed red arrows).

#### 2.3.1. Selection of Driver Variables

In order to produce the best fit between empirical data and the observable reality, the selection of the best set of model explanatory (or driver) variables for each transition is very important in urban LUC modeling [11]. While there are no universal set of driver variables for LUCs [44], several researchers have demonstrated that the selection of modeling drivers varies depending on the specific landscape changes in different study areas [11,19,20,45]. Although similar driver variables have been used in various studies, the degrees to which they contribute to LUCs differ [47]. Therefore, variables are selected based on study-area characteristics, observed LUCs, and expert knowledge of specific study areas.

In this study, we selected driver variables based on the premise that the expansion of intra-urban-LUs is driven by a combination of biophysical, social, economic, and policy/planning factors [48]. Reference was also made to two previous studies [8,29]. Simwanda and Murayama [8] investigated the spatial explanatory factors influencing the pattern of intra-urban-LU changes and also modeled the spatial dependency of unplanned and planned intra-urban-LUs in Lusaka. Simwanda, Murayama and Ranagalage [29] also investigated the drivers of intra-urban-LU changes in Lusaka using ANP modeling. This intensive approach involved the incorporation of ground surveys and inputs from a diverse set of experts.

Thus, based on expert knowledge of the study area and the observed intra-urban-LU changes, 13 driver variables were selected and used in this study (Figure 3 and Table 2). Table 2 outlines the selected drivers and their representation in terms of physical, neighborhood, social, economic, and policy/planning characteristics. The quality and availability of basic social, economic, and demographic data to inform policy continues to be a significant challenge across Africa [49]. Therefore, it is our argument in this study that scaling down to the intra-urban-LU level enables researchers to indirectly incorporate data that is otherwise not readily available or inaccurate for most SSA cities. In this study, for example, the Unplanned High-Density Residential (UHDR) class represents data on informal settlements (physical) with high population densities (neighborhood), high poverty levels (social/economic), and poor access to basic public services (policy and planning challenges). This kind of data is very challenging to obtain in most SSA countries.

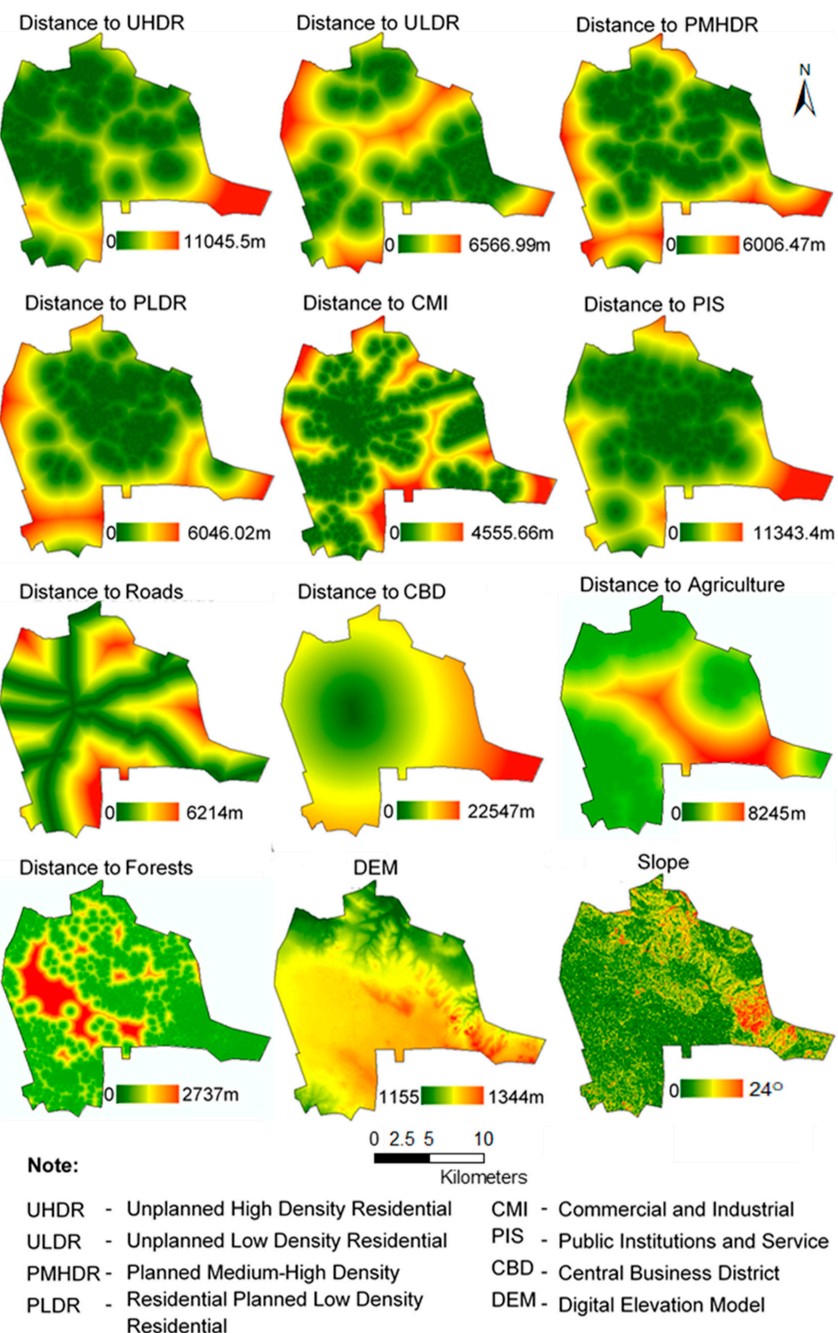

**Figure 3.** Model driver variables for the intra-urban land-use expansion of Lusaka city for the period.

### 2.3.2. Transition-Potential Modeling and Simulation

To carry out transition-potential modeling, we used the multilayer perceptron (MLP), a form of ANN that uses the back-propagation algorithm to calculate LUC transition potentials. The multilayer perceptron neural network (MLP-NN) model is one of the commonly used forms of ANN in urban LUC modeling [11,45,50]. The MLP-NN comprises one input layer, one or more hidden layers, and one output layer. When training the MLP-NN model, the input layer presents input data to the network. The input data is received by the hidden layers, which extract information from the input layers and express them as weights. The results in the output layer depend on the back-and-forth interactions among the input, hidden, and output layers.

According to Shafizadeh-Moghadam, Asghari, Tayyebi, and Taleai [51], the MLP-NN back-propagation algorithm starts with the ANN assigning random values to the network's weights and biases (ANN parameters), which is followed by the application of these random weights and biases to the input data by the MLP to estimate the outputs. Next, the ANN calculates the error between the estimated outputs and reference outputs, which is then propagated backward to the previous layers [51,52]. Based on the error-reduction principles and this back-and-forth iterative process, the weights and biases are updated until an acceptable error is obtained and the correct weights are assigned to the final output (for more details, see also [53]).

We combined the MLP-NN and Markov chain models provided in the LCM to create the neural network–Markov modeling framework. In the LCM, the MLP-NN is based on a multivariate function that integrates driver variables to predict the transition potential of LUCs between two time points at any location [11,54]. The Markov chain model computes transition rates over time [42]. The selected driver variables in Table 2 and intra-urban-LU maps were used to create submodels for each transition (i.e., nonbuilt to each intra-urban-LU class). In this study, calibration was done twice; intra-urban-LU maps for 2000 and 2005 were used to create transition submodels for simulation to 2010 (Figure 2, black arrows), while intra-urban-LU maps for 2005 and 2010 were used to create transition submodels for simulation to 2015 (Figure 2, dashed red arrows).

For each transition potential submodel (i.e., the change from nonbuilt to each of the six intra-urban-LU classes), we trained the MLP-NN using a random sample of 10,000 pixels, out of which 50% were used for training and 50% were used for testing. The MLP-NN was trained using three layers: the input and hidden layers with the same number of neurons or nodes as the selected number of driver variables (i.e., 13 variables; Table 2); and an output layer representing a transition potential map for each submodel. The other MLP-NN training parameters were set as 0.01 to 0.001 learning rate, 0.5 momentum factor, 1.0 sigmoid constant, and 0.01 acceptable error. The MLP-NN was run at 10,000 iterations for each transition using dynamic automatic training. The backward stepwise analysis in the LCM was used to remove the variables without power for each transition-potential submodel to reduce the likelihood of overfitting [42].

The accuracy values achieved for all transitions from nonbuilt to each of the six intra-urban-LU classes were acceptable, ranging from 87.87 % to 95.65 %. Eastman [54] recommended an accuracy rate of around 80%. Note that this accuracy is a measure of calibration and not validation [50]. We created transition-potential maps depicting the potential of each of the modeled transitions at each location within the study area. We then applied the Markov chain model to compute the transition probabilities expected in the future prediction based on the 2000 to 2005 and 2005 to 2010 intra-urban-LU changes. Finally, we simulated the intra-urban-LU maps for 2010 and 2015, and proceeded to model validation.

### 2.3.3. Model Validation

We used two indices for model validation: the Kappa index [55] and the figure of merit (FoM) [56]. Three alternative Kappa indices to the standard Kappa (Kno (Kappa for no ability), Klocation (Kappa for location), and Kquantity (Kappa for quantity)) were generated. Kno is an index that measures the overall agreement based on no information;

Klocation measures the agreement based on location only; and Kquantity measures the agreement based on quantity only [55,57]. Kappa indices range from −1 (indicating no agreement) to 1 (indicating perfect agreement) between two maps of the same variables.

However, the Kappa indices cannot provide components of agreement and disagreement between the observed and simulated intra-urban-LU maps. Pontius and Millones [58] presented that Kappa indices can be useless, misleading, and/or flawed for the practical applications in RS, and recommended using the cross-tabulation matrix to summarize simpler parameters, quantity disagreement, and allocation disagreement for accuracy assessment and map comparison. In this study, we cross-tabulated the observed and simulated 2010 and 2015 intra-urban-LU maps and computed four components of agreement and disagreement [59]: (1) observed change simulated correctly as change (hits); (2) observed persistence (i.e., all intra-urban-LU built-up class and non-built-up) simulated correctly as persistence (null successes); (3) observed change simulated incorrectly as persistence (misses); and (4) observed persistence simulated incorrectly as change (false alarms). Based on the hits, misses, and false alarms, the FoM was determined using Equation (1) [56]. In addition, three ratio indices (Equations (2)–(4)) that quantify the number of hits, misses, and false alarms relative to the observed change [60] were also determined.

$$\text{FoM} = \frac{H}{(H + M + F)} \quad (1)$$

$$\text{HOC} = \frac{H}{(H + M)} \quad (2)$$

$$\text{MOC} = \frac{M}{(H + M)} \quad (3)$$

$$\text{FOC} = \frac{F}{(H + M)} \quad (4)$$

where H, M, and F are the hits, misses, and false alarms, respectively, while HOC, MOC, and FOC are the respective ratios of hits, misses, and false alarms to the observed change (i.e., the sum of hits and misses), respectively.

### 2.3.4. Scenario Development for Future Intra-Urban-LU Simulation

Several studies have demonstrated that the development and simulation of future scenarios in urban studies are essential tools for future urban-development planning [11,17,22,60–63]. In this study, we developed three types of scenarios to simulate future intra-urban-LU expansion for the years 2020, 2025, and 2030: Scenario 1, business as usual or status quo; Scenario 2, environmental conservation and protection; and Scenario 3, strategic urban planning. Scenario 1, also referred to as a spontaneous scenario, was based on simulating future intra-urban-LU expansion following the historical land-transition trends without any constraints or modifications. Scenario 2 was also based on the historical land-transition trends but considered the full conservation and protection of all forests and other green spaces.

Lusaka, like many other SSA cities, has experienced rapid urbanization that has created pressure on the environment. High population growth coupled with increasing urbanization has outstripped land supply for development [64,65]. As a consequence, unplanned/informal settlements have increased, occupying ecologically valuable lands and impairing the provision of urban-ecosystem services [8]. Increased demand for social services (e.g., housing, water, and sanitation) has also drastically increased, creating pressure on the environment. Scenario 3 was developed based on this background.

Scenario 3 applied the same constraints as Scenario 2 (i.e., protection of green spaces and forests) but added a law that restricts the expansion of UHDR (i.e., informal settlements). Scenario 3 was developed based on the Lusaka urban-development plan (2010–2030; also referred to as the Lusaka Vision 2030), a policy proposal to restrict the expansion of informal settlements while upgrading or normalizing the existing ones, which are referred

to as statutory improvement areas [66]. To implement this policy, we added a constraint restricting the growth of the UHDR class from 2020 onwards. Under this scenario, all other residential land uses (i.e., ULDR, PMHDR, and PLDR) were allowed to expand to meet the housing demand. Of note is the assumption that, since the Lusaka urban development plan proposes to normalize all unplanned areas, the growth of ULDR areas was considered as planned expansion under Scenario 3. The purpose of Scenario 3 was to stimulate strategic sustainable urban-development planning while promoting environmental sustainability at the same time.

### 2.3.5. Intra-Urban-LU Change Detection and Analysis

After performing simulations of future intra-urban land-use expansion under the three scenarios, we then computed the area and percentage estimate of each intra-urban-LU class for each observed (2000, 2005, 2010, and 2015) and simulated (2020, 2025, and 2030) time point. The total area expansion of each intra-urban-LU class and its percentage contribution to the sum change of all intra-urban-LU classes were also computed for each of the time intervals and the temporal study extent. We then used the Pareto principle (also known as the 80/20 rule) to identify the intra-urban-LU classes that would significantly contribute to the total future intra-urban-LU expansion (i.e., sum change of all intra-urban-LU classes) under each scenario in the 2015–2020, 2020–2025 and 2025–2030 time intervals. The Pareto principle is a statistical distribution of data asserting that 80% of an outcome can be explained by 20% of the total observations [67]. In other words, the principle states that 80% of effects come from 20% of causes. Details of the Pareto principle can be found in [67,68]. In addition, we calculated the percent change of each intra-urban-LU class in each time interval using Equation (5) [8].

$$N_{ti} = \frac{(\text{size of net change of } i \text{ during } [Y_t, Y_{t+1}])100\%}{(\text{size of } i \text{ at time } Y_t)(\text{duration of } [Y_t, Y_{t+1}])}$$
$$= \frac{\left[\left(\sum_{j=1}^{J} C_{tji}\right) - \left(\sum_{j=1}^{J} C_{tij}\right)\right]100\%}{\left(\sum_{j=1}^{J} C_{tij}\right)(Y_{t+1} - Y_t)} \tag{5}$$

where $N_{ti}$ is the percent change for class $i$ during the time interval $[Y_t, Y_{t+1}]$ relative to the size of class $i$ at time point $t$; $t$ is the index for a time point; $Y_t$ is the year at time point $t$; $C_{tij}$ is the number of pixels that are class $i$ at time $t$ and class $j$ at time point $t + 1$; $i$ and $j$ are indices for an intra-urban-LU class, and $J$ is the number of intra-urban-LU classes. Note that the order of subscripts $j$ and $i$ in $C_{tji}$ in the numerator of Equation (1) is intentional so that the first summation computes the size of class $i$ at the final time $Y_{t+1}$.

## 3. Results

### 3.1. Intra-Urban-LU Maps and Expansion

Four intra-urban-LU maps were produced in this study (Figure 4), and Table 3 presents the intra-urban-LU expansion statistics for the years 2000, 2005, 2010, and 2015. The study revealed that intra-urban-LU expansion mainly resulted from the growth of UHDR, CMI, and ULDR areas during the entire study's temporal extent (2000–2015) (Table 3). UHDR areas expanded mainly to the south, west, and northwest, with some isolated expansion observed in the east and northeast. UHDR areas represented 29.8% of the total intra-urban-LU expansion (11,317 ha) during the temporal extent. CMI areas expanded mainly to the west of the CBD, representing 18.7% of the total intra-urban-LU expansion. ULDR areas had the third-largest expansion, primarily expanding in the southeast and accounting for 17.7 % of the total intra-urban-LU expansion. PLDR, PMHDR, and PIS accounted for 17.0%, 12.0%, and 4.9 % of the total intra-urban-LU expansion, respectively. Overall, residential intra-urban-LU expansion accounted for 76.4% of the total gain in intra-urban-LU, while CMI areas and PIS areas accounted for only 23.6%.

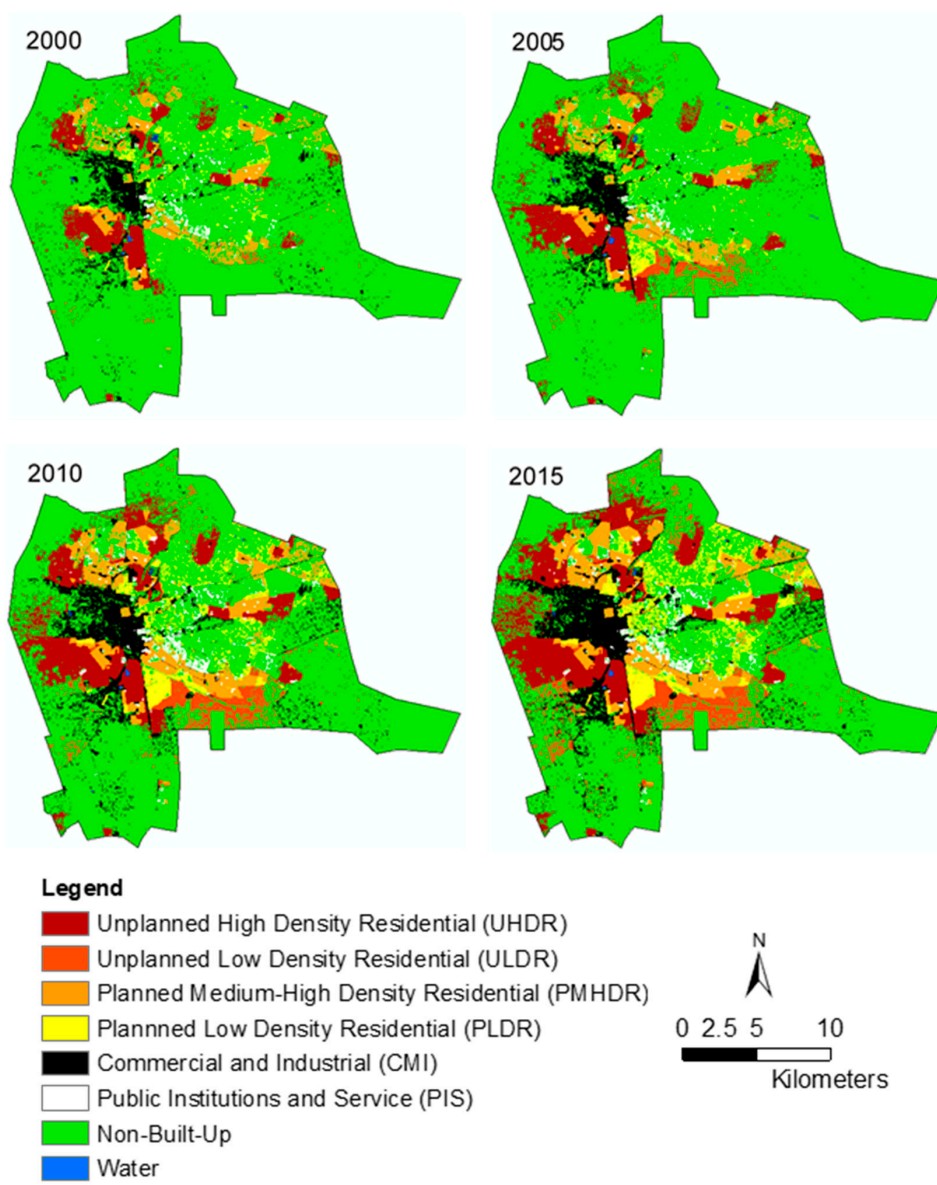

**Figure 4.** Urban-LU maps for Lusaka city, Zambia, for the years 2000, 2005, 2010, and 2015.

**Table 3.** Intra-urban-LU (ha) expansion statistics for Lusaka city, Zambia, in four time periods between 2000 and 2015.

| Urban-LU Class | Urban-LU | | | | | | | |
|---|---|---|---|---|---|---|---|---|
| | **2000** | | **2005** | | **2010** | | **2015** | |
| | **Area** | **%** | **Area** | **%** | **Area** | **%** | **Area** | **%** |
| UHDR | 2526 | 30.61 | 3520 | 33.57 | 4688 | 29.52 | 5898 | 30.14 |
| ULDR | 231 | 2.8 | 818 | 7.8 | 1589 | 10 | 2230 | 11.39 |
| PMHDR | 2196 | 26.61 | 2498 | 23.82 | 3217 | 20.26 | 3554 | 18.16 |
| PLDR | 900 | 10.91 | 1348 | 12.86 | 2061 | 12.98 | 2821 | 14.42 |
| CMI | 1931 | 23.39 | 2046 | 19.51 | 3365 | 21.19 | 4048 | 20.68 |
| PIS | 470 | 5.69 | 632 | 6.03 | 961 | 6.05 | 1020 | 5.21 |
| | 8254 | | 10,861 | | 15,881 | | 19,572 | |

**Table 3.** *Cont.*

| Urban-LU Class | Urban-LU Expansion | | | | | | | |
|---|---|---|---|---|---|---|---|---|
| | 2000–2005 | | 2005–2010 | | 2010–2015 | | 2000–2015 | |
| | Area | % | Area | % | Area | % | Area | % |
| UHDR | 993 | 38.11 | 1168 | 23.28 | 1210 | 32.78 | 3372 | 29.79 |
| ULDR | 587 | 22.5 | 771 | 15.35 | 641 | 17.37 | 1998 | 17.66 |
| PMHDR | 302 | 11.57 | 719 | 14.33 | 337 | 9.14 | 1358 | 12 |
| PLDR | 448 | 17.17 | 713 | 14.2 | 761 | 20.6 | 1921 | 16.97 |
| CMI | 115 | 4.42 | 1319 | 26.28 | 684 | 18.52 | 2118 | 18.71 |
| PIS | 163 | 6.24 | 329 | 6.56 | 59 | 1.59 | 550 | 4.86 |
| | 2607 | 100.0 | 5020 | 100.0 | 3691 | 100 | 11,318 | 100.0 |

*3.2. Model Validation*

Figure 5 shows the observed and simulated intra-urban-LU maps for 2010 and 2015, as well as their validation maps showing the location of the components of agreement and disagreement (i.e., hits, null successes, misses, and false alarms). The model-validation statistics are presented in Table 4. All Kappa index values were higher than 80%, suggesting good agreement between the observed and simulated 2010 and 2015 intra-urban-LU maps. The model-validation results also revealed that the simulated 2010 and 2015 intra-urban-LU maps had acceptable null successes, hits, misses, and false alarms relative to the observed intra-urban-LU maps for 2010 and 2015.

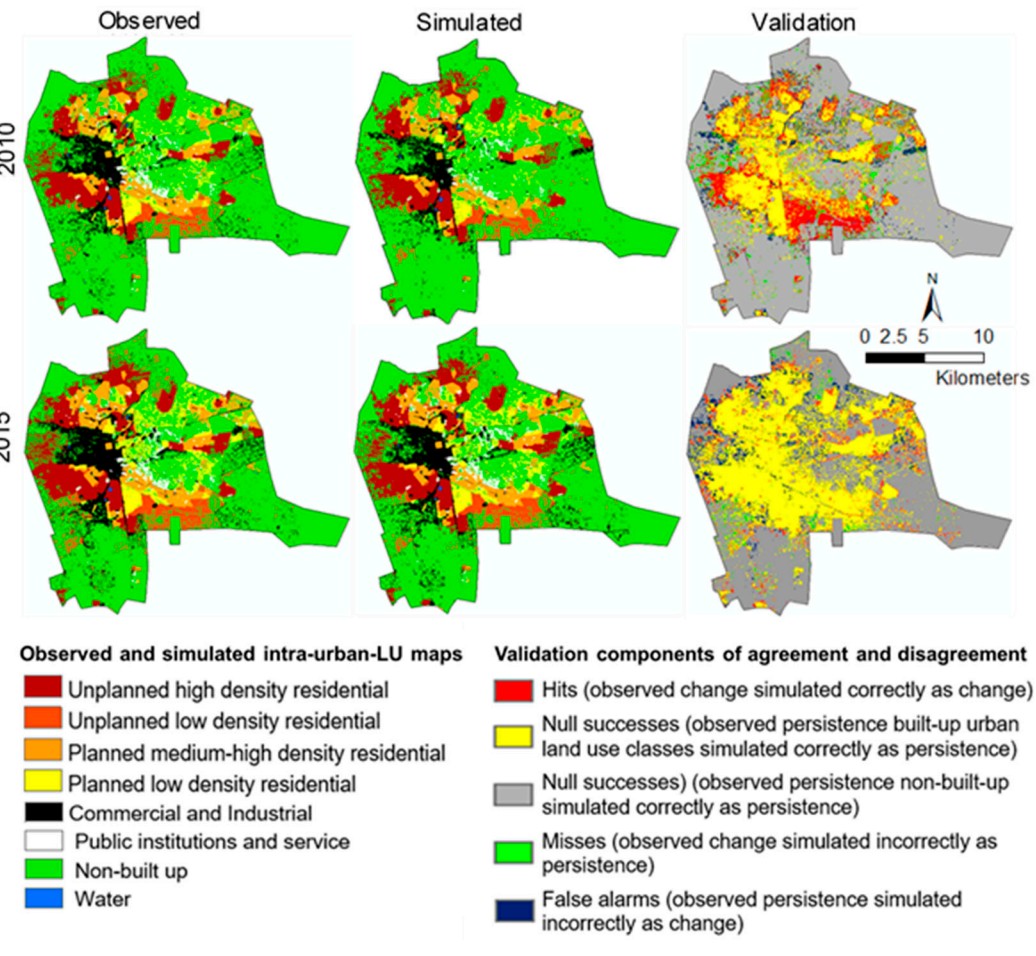

**Figure 5.** Observed and simulated intra-urban-LU maps for Lusaka city, Zambia, for 2010 and 2015 and their validation components of agreement and disagreement.

**Table 4.** Model validation results of observed and simulated 2010 and 2015 intra-urban-LU maps for Lusaka city, Zambia.

| Intra-Urban LU Map | Kappa Index (%) | | | Agreement and Disagreement (%) | | | | Ratio Indices | | | FoM (%) |
|---|---|---|---|---|---|---|---|---|---|---|---|
| | Kno | Klocation | Kquantity | N | H | M | F | HOC | MOC | FOC | |
| Simulated 2010 | 95.15 | 85.02 | 83.11 | 77.17 | 11.68 | 6.04 | 5.11 | 0.66 | 0.34 | 0.30 | 51.14 |
| Simulated 2015 | 94.05 | 92.62 | 91.09 | 88.46 | 8.40 | 1.34 | 1.80 | 0.86 | 0.14 | 0.19 | 72.76 |

Note: N, H, M, and F are null successes, hits, misses, and false alarms, respectively; while HOC, MOC, and FOC are the respective ratios of hits, misses, and false alarms to the observed change (i.e., the sum of hits and misses).

### 3.3. Scenario-Based Intra-Urban-LU Simulation

Figure 6 shows the spatial patterns of the simulated intra-urban-LU expansion in the three scenarios for the years 2020, 2025, and 2030. The intra-urban-LU expansion statistics for each scenario in each time interval are presented in Table 5. Figure 7 presents the Pareto analysis charts identifying intra-urban-LU classes that would significantly contribute to the total future intra-urban-LU expansion under each scenario. The results reveal that under business as usual (Scenario 1), over 80% of the intra-urban-LU expansion has been dominated by the growth of UHDR, followed by CMI and PMHDR areas during the 2015–2020 time interval. In the 2020–2025 and 2025–2030 time intervals, the intra-urban-LU expansion will still be dominated by UHDR and CMI, but PMHDR will be outpaced by PLDR areas by 2030 (Figure 7a–c). In terms of percent change during the entire 2015–2030 period, PIS is predicted to have the highest percent change (56%), followed by PLDR (47%), CMI (43%), UHDR (42%), PMHDR (29%), and ULDR (26%) (Figure 8) (see Equation (5).

**Table 5.** Intra-urban-LU (ha) scenario (S) simulation results for Lusaka city, Zambia, with projected results for 2025 and 2030. Refer to Figure 6 for abbreviations.

| Intra-Urban-LU Classes | Observed | | Simulated | | | | | | | |
|---|---|---|---|---|---|---|---|---|---|---|
| | 2015 | 2020 | | 2025 | | | 2030 | | | |
| | | S1 | S2 | S1 | S2 | S3 | S1 | S2 | S3 | |
| UHDR | 5898 | 6750 | 6747 | 7701 | 7694 | 6750 | 8359 | 8057 | 7035 | |
| ULDR | 2230 | 2230 | 2229 | 2524 | 2230 | 2230 | 2805 | 2523 | 2235 | |
| PMHDR | 3554 | 4102 | 4095 | 4482 | 4102 | 4102 | 4597 | 4266 | 4237 | |
| PLDR | 2821 | 3194 | 3192 | 3729 | 3515 | 3376 | 4158 | 3875 | 5817 | |
| CMI | 4048 | 4624 | 4614 | 5185 | 4973 | 4896 | 5785 | 5564 | 7690 | |
| PIS | 1020 | 1280 | 1280 | 1438 | 1280 | 1280 | 1588 | 1437 | 1310 | |
| Total | 19,571 | 22,181 | 22,157 | 25,059 | 24,959 | 22,635 | 27,292 | 23,957 | 28,325 | |
| Percent of total intra-urban-LU Expansion | | | | | | | | | | |
| UHDR | 30.1 | 30.4 | 30.5 | 30.7 | 30.8 | 29.8 | 30.6 | 32.8 | 24.8 | |
| ULDR | 11.4 | 10.1 | 10.1 | 10.1 | 8.9 | 9.9 | 10.3 | 9.3 | 7.9 | |
| PMHDR | 18.2 | 18.5 | 18.5 | 17.9 | 16.4 | 18.1 | 16.8 | 17.1 | 15 | |
| PLDR | 14.4 | 14.4 | 14.4 | 14.9 | 14.1 | 14.9 | 15.2 | 14.7 | 20.5 | |
| CMI | 20.7 | 20.8 | 20.8 | 20.7 | 19.9 | 21.6 | 21.2 | 20.8 | 27.2 | |
| PIS | 5.2 | 5.8 | 5.8 | 5.7 | 5.1 | 5.7 | 5.8 | 5.4 | 4.6 | |
| Intra-urban-LU Expansion | | | | | | | | | | |
| | 2015–2020 | | 2020–2025 | | 2025–2030 | | | 2015–2030 | | |
| | S1 | S2 | S1 | S2 | S1 | S2 | S3 | S1 | S2 | S3 |
| UHDR | 852 | 849 | 951 | 947 | 659 | 363 | 285 | 2461 | 2159 | 1137 |
| ULDR | 0 | 0 | 294 | 0 | 282 | 293 | 5 | 575 | 293 | 5 |
| PMHDR | 548 | 541 | 380 | 8 | 114 | 164 | 135 | 1043 | 712 | 683 |
| PLDR | 373 | 371 | 535 | 323 | 429 | 360 | 2441 | 1337 | 1054 | 2996 |
| CMI | 576 | 566 | 560 | 359 | 600 | 591 | 2794 | 1737 | 1516 | 3642 |
| PIS | 260 | 260 | 158 | 0 | 150 | 157 | 30 | 568 | 417 | 290 |
| Total | 2610 | 2587 | 2878 | 1637 | 2233 | 1928 | 5690 | 7721 | 6151 | 8754 |
| Percent Change | | | | | | | | | | |
| UHDR | 14.4 | 14.4 | 14.1 | 14 | 8.6 | 4.7 | 4.2 | 41.7 | 36.6 | 19.3 |
| ULDR | 0 | 0 | 13.2 | 0 | 11.2 | 13.2 | 0.2 | 25.8 | 13.1 | 0.2 |
| PMHDR | 15.4 | 15.2 | 9.3 | 0.2 | 2.5 | 4 | 3.3 | 29.3 | 20 | 19.2 |
| PLDR | 13.2 | 13.1 | 16.8 | 10.1 | 11.5 | 10.2 | 72.3 | 47.4 | 37.4 | 106.2 |
| CMI | 14.2 | 14 | 12.1 | 7.8 | 11.6 | 11.9 | 57.1 | 42.9 | 37.4 | 90 |
| PIS | 25.5 | 25.5 | 12.3 | 0 | 10.4 | 12.3 | 2.4 | 55.7 | 40.9 | 28.5 |

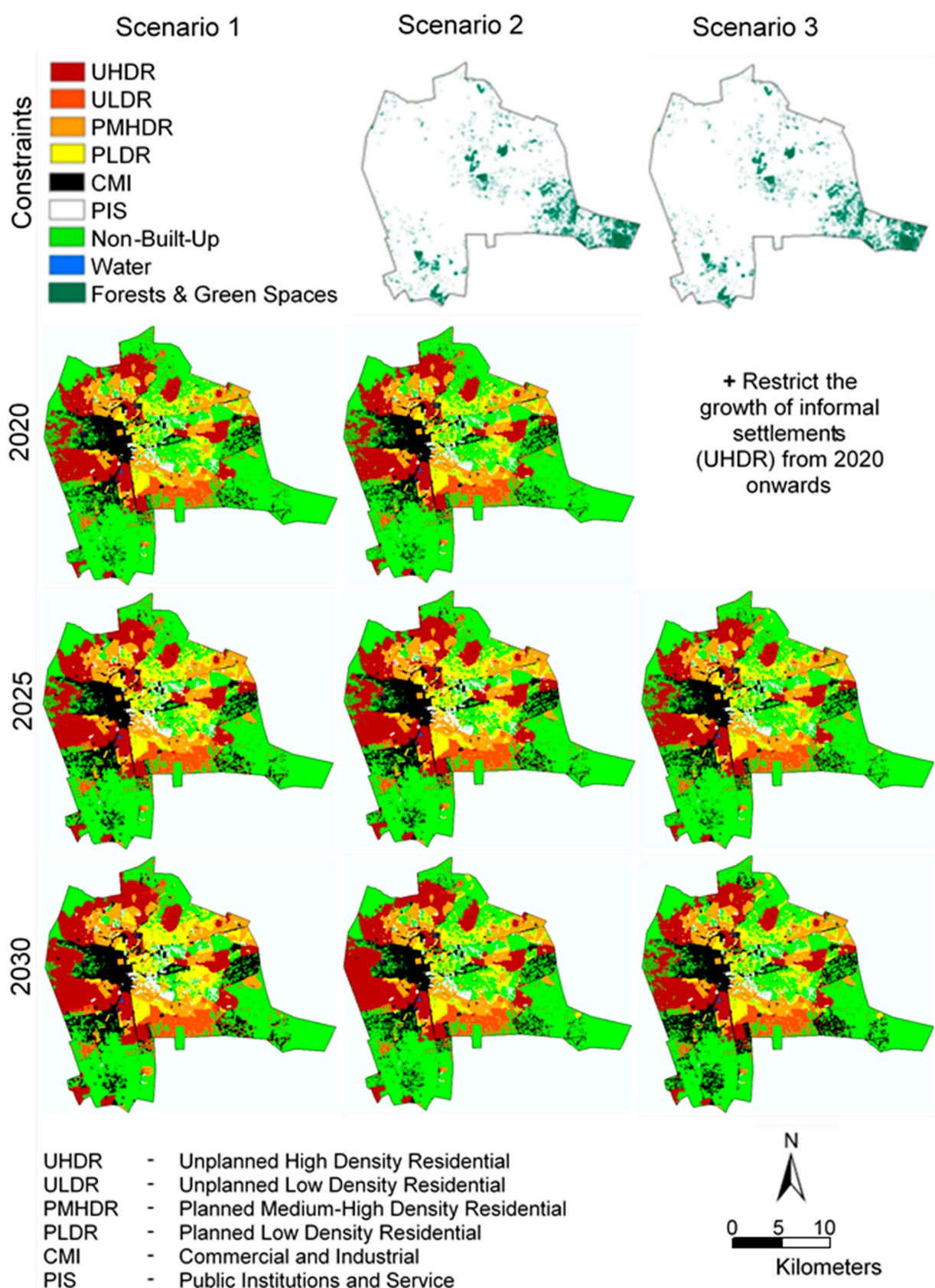

**Figure 6.** Simulated urban-LU maps for three scenarios in 2020, 2025, and 2030 for Lusaka city, Zambia: Scenario 1, business as usual; Scenario 2, environmental conservation and protection; and Scenario 3, strategic urban planning.

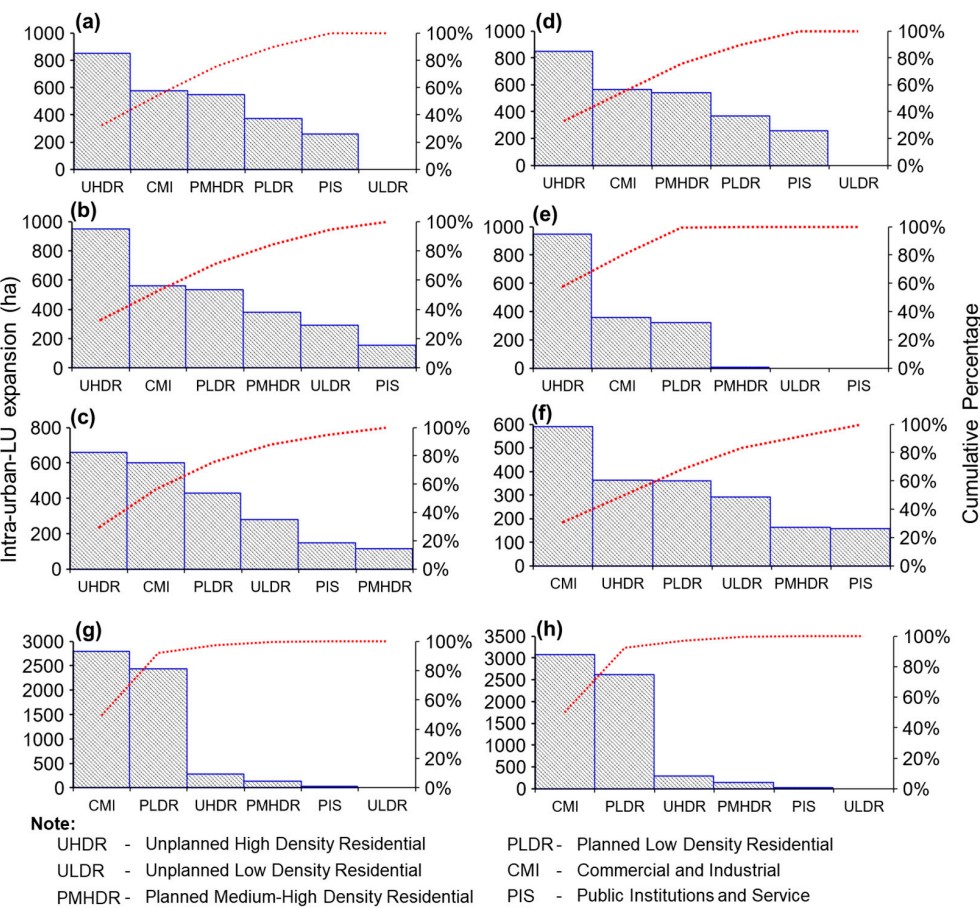

**Figure 7.** Pareto analysis charts identifying simulated urban-LU classes that would significantly contribute to the total urban-LU expansion for Lusaka city, Zambia. Scenario 1: (**a**) 2015–2020, (**b**) 2020–2025, (**c**) 2025–2030; Scenario 2: (**d**) 2015–2020, (**e**) 2020–2025, (**f**) 2025–2030; and Scenario 3: (**g**) 2025–2030 and (**h**) 2020–2030.

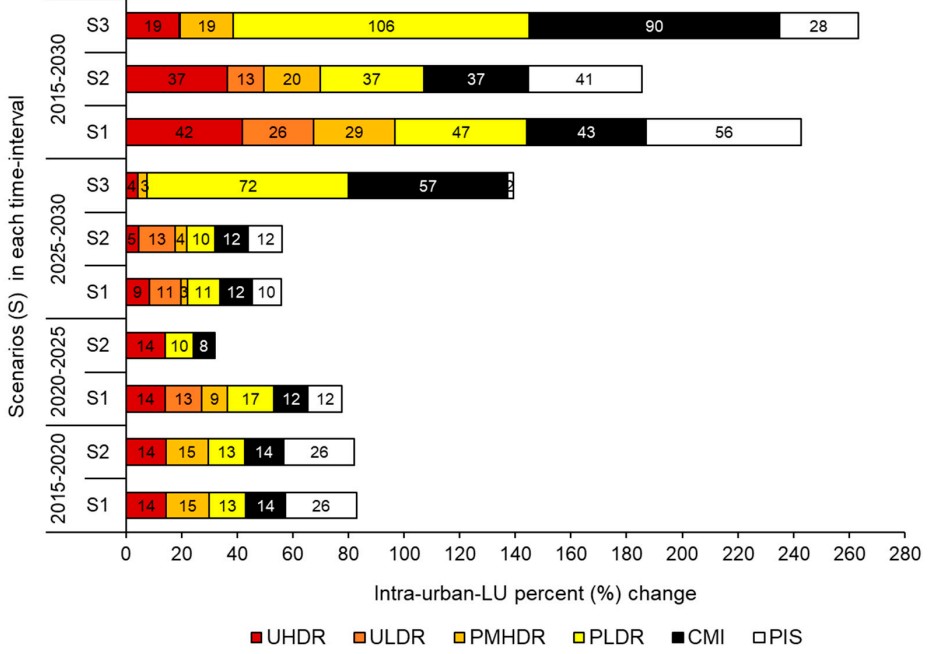

**Figure 8.** Percent change of each intra-urban-LU class for Lusaka city, Zambia, in each simulated time interval. Refer to Figure 7 for abbreviations.

Scenario 2 (environmental conservation and protection) showed minimal differences with Scenario 1 in the 2015–2020 time interval. Similarly, over 80% of the intra-urban-LU expansion resulted from the growth of UHDR areas, followed by CMI and PLDR areas under Scenario 2 (Figure 7d–f). However, Scenario 2 further predicts that with continued protection of green spaces and forests through the 2020–2025 and 2025–2030 time-intervals, the expansion of all intra-urban-LUs will be reduced. Particularly, the expansion of ULDR areas will be reduced to almost zero. Nevertheless, the expansion of UHDR areas will still be very dominant despite the expansion being reduced slightly (i.e., 37 % change) when compared to Scenario 1 (42% change) (Table 3). Under Scenario 2, the percent change of CMI and PLDR will be 37% each, while that of PIS will be 41% during the entire 2015–2030 time interval (Figure 8).

Conversely, under the strategic urban planning scenario (Scenario 3), constraining the expansion of UHDR areas from 2020 onwards, about 99% of intra-urban-LU expansion will be from CMI and PLDR areas by the year 2030 (Figure 7g,h). The expansion of UHDR and ULDR areas will be reduced significantly. During the 2025–2030 time interval, the percent change of UHDR areas will reduce to only 4.2%, while ULDR areas will only change by 0.2%. In both Scenario 2 and Scenario 3, the model predicts the relatively low expansion of PMHDR and PIS areas. In terms of percent change during the entire 2015–2030 period, PLDR (106%) and CMI (90%) are predicted to have the highest change, followed by PIS (28%), UHDR (19%), and ULDR (19%) (Figure 8).

## 4. Discussion

### 4.1. Intra-Urban-LU Expansion and its Drivers

In this study, we assessed the expansion of intra-urban-LUs in the rapidly urbanizing SSA city of Lusaka, Zambia from 2000–2015. The UHDR areas (i.e., informal settlements) dominated the expansion of all urban-LUs, contributing about 32.78% during the 2000–2015 study period, indicating a rapidly growing urban population in Lusaka. Census data for Lusaka shows an increase of 658,276 people from 2000 (1,084,703) to 2010 (1,742,979), compared to an increase of only 323,639 people between 1990 (761,064) and 2000 [40,41]. Lusaka accounts for at least 32% of the total urban population in Zambia [40,69]. The population in Lusaka is largely characterized by the urban poor, who have continuously demanded low-cost housing since independence [70]. The relatively slow expansion of PMHDR and PIS areas in this study suggests that government efforts to respond to the high demand for low-cost housing and subsequent public and social services have been overtaken by the rapidly growing population. Consequently, informal settlements have outpaced the expansion of all other urban-LUs in Lusaka.

Our results are consistent with the findings of [8,29]. Analogous to the observations in this study, Chitonge and Mfune [65] reported that by 2010, 65% of the population in Lusaka was distributed around 37 informal settlements. This situation is not unique to Lusaka or Zambia; Laros and Jones [71] also reported that about 60% of all urban dwellers in SSA reside in informal settlements or slums. Informal settlements in Lusaka and other SSA cities have many problems, including high urban poverty levels, lack of access to basic services (roads, clean water, reliable power, etc.), unemployment, and environmental pollution resulting from unplanned waste disposal and sewer discharge, as reported by several researchers [8,29,72–74].

The expansion of ULDR and PLDR areas was also relatively significant, contributing about 17.37% and 20.60%, respectively, to the total urban-LU expansion from 2000 to 2015. In the past, this has been attributed to intra-urban migration that led to an increase in the urban population demanding better housing than that found in informal settlements [8,75]. The privatization of the mining and related industries in other major urban cities in Zambia in the 1990s and 2000s has been the main driver for urban–urban migration [65]. The expansion of ULDR areas has also resulted from the lack of proper urban planning and development systems that have allowed the emergence of informal land markets and corrupt forms of land acquisition [8,65]. ULDR areas signal a problematic future, as they

also lack proper roads, experience flooding, have limited connection to municipal water and sanitation services, and are reported to occupy the remaining ecologically valuable lands in the periphery areas of the city [29].

On the other hand, the expansion of CMI areas accounted for about 18.52%, which also indicated a relatively substantial contribution to the total urban-LU expansion during the 2000–2015 study period. Simwanda and Murayama [8] modeled the spatial dependency of unplanned and planned intra-urban-LUs in Lusaka, and linked the growth of informal settlements to CMI areas. Therefore, it is plausible that the expansion of CMI areas in Lusaka, mainly comprising small and medium manufacturing enterprises and trading markets, has been driven by the availability of a large but cheap labor force coming from the informal settlements. Furthermore, many owners of commercial and industrial activities have also taken advantage of the informal land markets to establish new developments [29].

### 4.2. Modeling and Validation

In this study, we applied the neural network–Markov model to simulate scenarios of future intra-urban-LU expansion in Lusaka city, Zambia. Four urban-LU maps (2000, 2005, 2010, and 2015) and selected biophysical, neighborhood, social, and economic explanatory variables were used to calibrate and validate the model (see Section 3). We used two indices for model validation: the Kappa index (Kno, Klocation, and Kquantity) [55] and FoM [56]. The validation results showed Kappa index values greater than 80%, suggesting good agreement between the observed and simulated intra-urban-LU maps. Other researchers scaling down to intra-urban level have reported somewhat similar high Kappa values [20,22]. However, other scholars have criticized Kappa for giving high accuracy values in study areas with few LUC changes, and vice versa [76,77]. Scaling down to intra-urban-LUs reduces the quantity of LUC changes to each intra-urban-LU class, and may show high Kappa values compared to one simple individual urban-LU class (i.e., built-up).

Therefore, to verify the simulation accuracy of the model, we employed the FoM [20,22,45,46]. The FoM is a ratio metric that is unaffected by the quantity of LUC changes [45]. We found FoM values of 51.14% and 72.76% for the first- and second-round simulations of the 2010 and 2015 intra-urban-LU maps, respectively (see Section 3 and Figure 2). The FoM values in the literature have generally ranged from 1% to 59% based on varying study area spatial extents and imagery spatial resolutions, as well as the number of land-use/land-cover classes [20,45,46,56]. Our results for the second-round simulation to 2015 were slightly higher than previous studies. Therefore, the simulation accuracy of the neural networks Markov model in this study is acceptable. Thus, this study has proven that the neural-network–Markov model can simulate the complex non-linear and unbalanced mixed development of unplanned and planned intra-urban-LUs in SSA cities and other regions with similar development patterns.

### 4.3. Scenario-Based Intra-Urban-LU Simulation

After model validation and establishing satisfactory reliability, the neural network–Markov model was used to simulate urban-LU expansion for three scenarios (business as usual/status quo, environmental conservation and protection, and strategic urban planning) for the years 2020, 2025, and 2030. The results suggest that a business-as-usual scenario (Scenario 1) is perilous. Without any policy interventions in the observed past (2000–2015) and future (2020–2030) urban-LU expansion trends, the business-as-usual scenario signals an escalating problem of unplanned development. The informal settlements or shanty compounds (herein referred to as UHDR) and other unplanned residential extensions (herein referred to as ULDR) are predicted to outpace all other urban-LUs by 2030. These unplanned developments are predicted to largely occupy the remaining ecologically valuable lands in the periphery areas of Lusaka. As a consequence, the already-impaired provision of urban-ecosystem services will be lessened further. The expansion of unplanned areas will also exacerbate environmental pollution as these areas are characterized by unsafe water abstraction and sanitation activities, indiscriminate waste disposal, and frequent flooding [78].

It also means that, by 2030, Lusaka will have millions of more people living in these informal settlements than the recently predicted 65% [65,69]. Thus, Scenario 1 is unsustainable.

By contrast, under the environmental conservation and protection scenario (Scenario 2), the disturbance of the remaining green spaces and forests will be much less. This fact will promote the restoration of the urban ecosystem. However, given that most of the green spaces and forests have been depleted and are observed to exist mainly in the periphery areas, achieving substantial urban ecosystem restoration is uncertain. This is evidenced by the results of this study showing no significant differences between Scenarios 1 and 2. Although Scenario 2 shows that the expansion of unplanned low-density residential shall be under control by 2030, the expansion of informal settlements will occur almost at the same rate as in Scenario 1. The model predicts that the nongreen open spaces in the periphery areas will still be potentially invaded by informal settlements. This means that the pressures described in Scenario 1 above will still occur.

On the other hand, under the scenario of strategic urban planning (Scenario 3), it was observed that unplanned expansion would be brought under control while protecting the green spaces and forests at the same time. Therefore, Scenario 3 appears to be a more desirable way forward, as it has the potential to achieve both environmental and urban sustainability. However, while the unplanned expansion might be controlled under Scenario 3, we also observed that about 99% of intra-urban-LU expansion would be from CMI and PLDR areas by the year 2030. The CMIs areas are predicted to occupy the remaining unoccupied land both within and in the periphery areas of the city. The PLDR areas will potentially become medium to high density, as there is an infill expansion predicted to occupy the open green spaces (e.g., backyards and parks) within these areas. This may still impede environmental and urban sustainability.

The observations from our scenario-based intra-urban-LU simulation are comparable to other studies. Similar to our observations, previous scenario-based urban simulation studies have stressed that the business-as-usual scenario is very unsustainable [11,17,60,79]. Unlike our study area, where the environmental-protection scenario (Scenario 2) alone was proven insufficient for attaining urban sustainability, other studies (e.g., [11,60]) found this same scenario to be attainable. The major difference was mainly because the other cities have a substantial amount of green spaces, croplands, and forest areas that can be used for the restoration of the urban areas, a situation which is not obtainable in Lusaka. The strategic-planning scenario (Scenario 3) can be compared to the restrictive/simulative and punitive scenarios presented in [17,61], respectively. These scenarios assume the adoption of policies that prevent the growth of certain developments (e.g., informal settlements in this study) in the undesired zones. Similar to this study, [17] and [61] discuss the implications (see Section 4.4) of the proposed policies, and recommend strategic planning for the attainment of their respective goals. The different scenarios offer guidance in taking strategic policy directions and setting up a planning vision [17].

### 4.4. Implications for Sustainable Urban Landscape Planning and Development

Based on the scenarios presented in this study, attaining Scenario 3 seems desirable for achieving sustainable future urban-development planning. Attaining Scenario 3 is also in line with the UN's Global SDGs, particularly SDG No. 11 (Sustainable Cities and Communities), which aims to make cities and human settlements inclusive, safe, resilient, and sustainable by 2030 [80]. Therefore, its implications should be considered.

Environmental sustainability is arguably attainable under Scenario 3. This can be achieved by combining the assumed protection of existing green spaces and forests with activities such as urban regreening and vegetation rehabilitation. This study also observed that in the three scenarios, but especially in Scenario 3, the need for commercial and industrial expansion (herein referred to as CMI) is predicted to be relatively significant up to 2030. Therefore, we suggest a deliberate policy to set up satellite economic zones around the remaining available land in the periphery areas of the city as well as the adjacent districts, which are predominantly rural. Satellite economic zones have the potential to decongest the

city, which will provide an opportunity to redesign the city and restore urban-ecosystem services, thereby promoting environmental sustainability. Urban ecosystem services can be restored from a diverse set of habitats, including green spaces such as parks, cemeteries, vacant lots, gardens and yards, campus areas, and blue spaces (mainly streams, ponds, and dams) [81].

With that stated, it is irrefutable that environmental sustainability can only be achieved if it goes hand-in-hand with the nexus of social and economic sustainability. In fact, urban sustainability is often recognized as a concept comprising environmental, social, and economic elements [82,83]. Therefore, while the restriction of the expansion of unplanned/informal settlements is crucial to achieving urban sustainability under Scenario 3, the social and economic implications of this approach need consideration. Perhaps some important questions will need to be answered if Scenario 3 or urban sustainability is to be attained in Lusaka city. How will the city meet the high demand for housing, especially for the urban poor? What strategies are required to avoid the re-emergence of unplanned settlements? How will the influence of other forces (e.g., economic development and political forces) be managed? Thus, the implementation of Scenario 3 should be more strategic rather than completely restrictive. Kamete [84] warned that focusing on punitive and rigid planning enforcement laws in Africa has proven to be ineffective and in some cases counter-productive, particularly when dealing with informal settlements.

A concerted effort engaging key players including, but not limited to, urban planners and policymakers, researchers, technocrats, local urban dwellers in Lusaka city, and players in adjoining exurban areas that support urban sprawl is required. We acknowledge that the answers to how urban sustainability can be achieved in Lusaka city are not easily acquired. However, the questions raised above provide guidance and could stimulate visionary debate and urban planning that can create a clear road map to sustainability, as is the intention of the scenario analysis presented in this study.

## 5. Conclusions

In this study, we examined real (2000–2015) and simulated future (2020–2030) scenarios of intra-urban-LU expansion in the rapidly urbanizing SSA city of Lusaka, Zambia. The results revealed that Lusaka city has experienced rapid urban expansion dominated by informal settlements and that rapid urbanization threatened urban sustainability in the city.

To support efforts aimed at attaining urban sustainability, this study developed a neural network–Markov model to simulate three scenarios of future urban-LU expansion in Lusaka (business as usual/status quo, environmental conservation and protection, and strategic urban planning) up to 2030 in line with national and global sustainable development goals. The results suggested that a business-as-usual setup is perilous, as it signals an escalating problem of unplanned settlements. The environmental conservation and protection scenario is insufficient to achieve urban sustainability, as most of the green spaces and forests have been depleted. The strategic urban planning scenario has the potential for achieving sustainable urban development, as it predicts sufficient control of unplanned settlement expansion and protection of green spaces and forests.

Our study recommends several strategies, including the provision of planned high-density housing for the urban poor to avoid re-emergence of informal settlements, improvement of land-tenure policies and delivery systems, the establishment of satellite economic zones to decongest the city, investment in both green and blue infrastructure, and timely policy reviews. In summary, this study provides guidance and attempts to stimulate visionary debate and urban development planning that can create a clear road map to sustainability. The study is a contribution to LUC change-modeling and simulation studies that serve as early-warning tools. The information provided can also be beneficial to other African cities and other developing regions experiencing similar urban-LU expansion, especially in consideration of unplanned versus planned expansion.

For future research, the application of convolutional neural networks (CNNs) in place of the MLP-NN is recommended. Unlike MLP-NN, CNNs can take into account neighborhood

characteristics by exploiting the spatial or temporal correlation of pixels in the input intra-urban-LU images. This will allow researchers to investigate, for example, the variations of error across the study area or particular events in specific years (man-made or natural) that could have influenced how the unplanned or planned areas expanded but could not be predicted by the model. The study also considered the commercial and industrial intra-urban-LUs as an individual intra-urban-LU class. Further studies should consider separating the commercial and industrial intra-urban-LUs for better insights on their growth and overall future urban planning.

**Author Contributions:** Conceptualization, M.S. and Y.M.; data curation, M.S.; formal analysis, M.S.; methodology, M.S, and Y.M.; project administration, Y.M.; software, M.S. and Y.M.; supervision, Y.M.; writing—original draft, M.S.; writing—review and editing, Y.M., D.P., V.R.N., and M.R. All authors have read and agreed to the published version of the manuscript.

**Funding:** This study was supported by the Monbukagakusho scholarship program of Japan for post-graduate studies and the JSPS grant 18H00763 (2018–20).

**Institutional Review Board Statement:** Not applicable.

**Informed Consent Statement:** Not applicable.

**Data Availability Statement:** No new data were created or analyzed in this study. Data sharing is not applicable to this article.

**Acknowledgments:** The authors are grateful to the editor and the anonymous reviewers for their helpful comments and suggestions to improve the quality of this paper.

**Conflicts of Interest:** The authors declare no conflict of interest.

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
