# Peer review of "Simulating Scenarios of Future Intra-Urban Land-Use Expansion Based on the Neural Network–Markov Model: A Case Study of Lusaka, Zambia"

_remotesensing, doi:10.3390/rs13050942_

Round 1
Reviewer 1 Report
- his study uses data derived from RS and 133 GIS techniques including intra-urban-LU maps (2000, 2005, 2010, 2015), and selected bio-physical, neighborhood, social, and economic explanatory variables to develop, calibrateand validate a Neural Network-Markov model to simulate the expansion of unplanned and planned intra-urban-LUs in Lusaka, Zambia (2000 - 2030).
Reference is made to the paper [38] in which the results obtained for the years 1990-2010 have better accuracy. Why not make a comparison between the quality of the results obtained by Neural Network-Markov model for the period 2000-2010 with those obtained previously.
- The intra-urban-LU maps for 2000 and 2010 were obtained from [38] while the intra-urban-LU maps for 2005 and 2015 were processed in this study. The methods are different and a comparison should have been made between the accuracies obtained to highlight the questionable progress of the current method.
- In Figure 5 - Observed and simulated intra-urban-LU maps for Lusaka city, Zambia, for 2010 and 414 2015 and their validation components of agreement and disagreement – it is noted that for 2015 no notable changes are validated.
Author Response
Please find attached our responses

Reviewer 2 Report
The manuscript provides an insight on what can be predicted about land use change in the sub-Saharan region of Africa, considering the capital city of Lusaka as a case study, by using neural networks and Markov models from data derived from GIS and remote sensing sources. The work is certainly of importance for providing guidance while planning land use and attain better sustainability by 2030. However, my suggestion is that there are a few elements that need to be justified and analyzed better before publishing the paper.
The manuscript is clear, well written and easy to follow.
I think the abstract and keywords provides a good summary of the topic being covered.
The introduction is generally well written. There are a few remarks to be made, although most of them are of minor importance. Some references are used more than once. While this might happen (for example a reference can be needed in different contexts sentences or pages apart) a few of them are used a second time after just a few rows (examples are lines 37-41 or 48-49). In these cases they might just be redundant. Moreover, some sentences lack a space between them (examples in lines 76, 81). Some references, provided together are out of order, with no apparent reason and this might hinder readability (example in lines 72, 87). Also regarding citations, sometimes multiple styles are used at the same time (see line 132).
However, regarding the introduction part, and this might extend its importance to the rest of the paper too, while it is clear that there are differences between the works presented in the introduction and the work presented in the paper, a few additional references, with different kind of connections seem to be missing. While “googling” on scholar for “neural network markov land use” some titles that attracted attention were “Predicting the trend of land use changes using artificial neural network and markov chain model (case study: Kermanshah City)”, “Land use change modeling through an integrated multi-layer perceptron neural network and Markov chain analysis (case study: Arasbaran region, Iran)”, “A remote sensing aided multi-layer perceptron-Markov chain analysis for land use and land cover change prediction in Patna district (Bihar), India”. While I understand that, as reported, none of these works concern SSA cities, what it does not establish convincingly are the particular characteristics of the area, or the kind of study done, that do not allow to build on the work already done or to compare with it later, even considering some differences that certainly apply.
In the materials and methods section, the study area is covered with sufficient detail from a geographic point of view. Data and intra-urban-LU classification is also stated clearly. However, while I understand the degree of granularity needs to be kept low, I do not understand why commercial and industrial activities are considered together, since, in my opinion, their land use need (and the way they impact human life) are quite different. I understand that this might be covered in the related paper [38], but it is important to state those choices here as well.
Section 2.3 is mostly explained well. There are, however, some details that should be clarified. In section 2.3.2 MLP-NN are explained. I suggest to cite directly the author of the back-propagation, in addition to the review article. “Learning representations by back-propagating errors”, https://www.nature.com/articles/323533a0#citeas. Even more important, is that while the authors are able to explain the way mlp-nn can be trained with supervised learning, the explanation ends there. Details related to the network employed in their work are completely missing, starting from explaining the number of inner layers used during the study, along with other data that would enable other people to repeat the experiment (number or neurons, activation functions, and so on) or at least understand the choices that have been made. It is also puzzling, as explained before, that other works that seem very similar, made in the past, are not cited, nor compared with. Other information seem missing as well: in the sentence starting at line 290 it is explained the way the network has been trained. It explains that 10.000 pixels have been used in total, but starting from what? The sampling is really random? No better way to sample the territory? Maybe the authors made the right choice, but should explain it better, so that the reader does not have the same doubts. Citation styles is still inconsistent, now and then (example line 356).
Formatting and citation issues are common in Section 3 as well. While evaluating results it would also be beneficial to have a comment about them, trying to explain why the simulation deviated from reality, even if they are just possible explanations, to be confirmed by a future study, adding some details in the process. For example, the error is uniform across all districts or not? Can it be explained? Have particular events in particular years (man-made or natural) altered in a significant way the city that could not been predicted/learned by the system?
Section 4 and 5 are mostly fine, with just a few issues (I do not understand “a situation which is not obtaining in Lusaka”, line 587). Please check citation styles anyway. Speaking of the results, it looks like most of the time they concur with what was already found in previous work. If there are some new insights, where the proposed methodology shows its strength, please highlight them better.
Regarding bibliography, some of the entries seem incomplete (like [39]). Please check carefully.
Author Response
Please find attached our responses

Reviewer 3 Report
Urbanization is indeed a global phenomenon with strong sustainability implications across multiple scales. As one of the most significant signs of urbanization, the expansion of urban land has attracted plenty of academic attentions. Studies in the Global South and particularly in Africa are at the forefront given the limited availability of data resources and technology. Taking Lusaka in Zambia as an example, this work in general is a timely contribution to the research on urban sustainability practices in Africa and can for sure result in a better understanding of the trajectory of urbanization in SSA cities and its impacts. Furthermore, the future simulation at three different scenarios may have potential in guiding the sustainability of urban Africa. Methodologically, this work tries to advance the simulating approach of by scaling down to the intra-urban-LU level, which is also a potential contribution to the field of LUCC.
In a nutshell, the manuscript is well-organized and easy-to-follow. I therefore suggested an accept subject to some minor revisions.
1) Section 4.4, the authors proposal more specific policy suggestions regarding the future urban planning by considering not only the simulation results but also the driving mechanisms.
2) Figure 3, the unit in the map of Slope says m (meter)?
3) The authors may want to discuss more theoretical implications for the advancement of urban theories in Africa and beyond.
Author Response
Please find attached our responses

Round 2
Reviewer 1 Report
1. It was not specified how the MLP neural network training was validated.
2. No criteria used to stop MLP neural training have been specified
3. No comparison was made with previous reference results [38-
Author Response
Please find attached our responses

Reviewer 2 Report
The reviewer appreciates the improvements to the manuscript.
In particular, comments no. 1, 2 and 6 are properly and satisfactorily addressed (some minor citation issues remain, like line 1054). Authors in comment 6 also rightly point out that some analysis can be postponed in a future work, as long as this is noted in the conclusions.
Regarding comment 7 (part1), although the reviewer is not a native English speaker, "situation which is not obtaining in Lusaka" seems wrong. Please check the grammar (the explanation that followed is better); "situation which is not possible in Lusaka"/"situation which is not obtainable in Lusaka" might work as well.
Comment 3 is also addressed better, citing some of the references that have been provided, such as "A remote sensing aided multi-layer perceptron-Markov chain analysis for land use and land cover change prediction in Patna district (Bihar), India", which fittingly also uses NN+Markov models included in LCM.
Comment 4 has not been addressed. The reviewer understands that the authors provide better insights in LUC by distinguishing between residential/commercial/public services where others handle them as a whole. The authors approach is commendable and agreeable. However, the reviewer points out that some economic activities should not take place near one another: a coal mine or a steel factory should not be placed near a market. Considering them together, in reviewer's opinion, seems like something that should be avoided, for a better planning. It is understandable that this might require further work, but it can be at least be noted as a possible area to investigate.
Comment 5 has been partially addressed. It is worth noting that the comment did not want a complete explanation of LCM. The comment wants to make the experiments "repeatable", by asking to provide the parameters that have been used for the neural network. While some of them are reported (such as learning rate) others are missing or difficult to retrieve from the text. Just for comparison, one of the provided citations (https://maxwellsci.com/msproof.php?doi=rjees.6.5763) show them together in Table 1 (page 218). There are also "signs" that the comment was not correctly understood or there is some confusion between nodes and layers. In the response given:"Kindly note that various studies have chosen varying numbers on input and hidden layers"... This is wrong. There is just one input layer and one output layer in a MLP NN. However, there might be multiple hidden layers, each one with different number of nodes. Do the authors use just one hidden layer? with how many nodes in the hidden layer? It is understandable that LCM might hide some choices and complexities but it is highly improbable that such basic information cannot be extracted or set.
Regarding Comment 7 part2 ("Speaking of the results, it looks like most of the time they concur with what was already found in previous work") it should be pointed out that "most" is different from "always". Probably it sounded too harsh but there is hope the comment provided a way for authors to remark and explain better the key insights.
The reviewer would also suggest the authors consider convolutional neural networks, instead of MLP NN, with Markov models as possible future investigations. Cellular Automata, as noted by the authors, have their shortcomings but at least CA tries to model the evolution of something by considering neighbors. Convolutional neural networks do a better job in considering "neighbors" than normal MLP NN.
Please note that the overall recommendation of "accept after minor revision" is given considering that the manuscript is very polished in some aspects but can be improved in others. Therefore it is strongly recommended that the conclusions are expanded so that authors explain that some points (reported above) might be addressed (or addressed better) in a future work.
Author Response
Please find attached our responses
